# Actin Cytoskeleton Straddling the Immunological Synapse between Cytotoxic Lymphocytes and Cancer Cells

**DOI:** 10.3390/cells8050463

**Published:** 2019-05-16

**Authors:** Hannah Wurzer, Céline Hoffmann, Antoun Al Absi, Clément Thomas

**Affiliations:** 1Cytoskeleton and Cancer Progression; Laboratory of Experimental Cancer Research, Department of Oncology 84 Val Fleuri, L-1526 Luxembourg City, Luxembourg; hannah.wurzer@lih.lu (H.W.); celine.hoffmann@lih.lu (C.H.); antoun.alabsi@gmail.com (A.A.A.); 2University of Luxembourg, Faculty of Science, Technology and Communication, 2 Avenue de l’Université, L-4365 Esch-sur-Alzette, Luxembourg; 3University of Strasbourg, 67081 Strasbourg, France

**Keywords:** actin cytoskeleton, cytotoxic T lymphocytes, immune evasion, immune surveillance, immunological synapse, natural killer cells

## Abstract

The immune system is a fundamental part of the tumor microenvironment. In particular, cytotoxic lymphocytes, such as cytolytic T cells and natural killer cells, control tumor growth and disease progression by interacting and eliminating tumor cells. The actin cytoskeleton of cytotoxic lymphocytes engaged in an immunological synapse has received considerable research attention. It has been recognized as a central mediator of the formation and maturation of the immunological synapse, and its signaling and cytolytic activities. In comparison, fewer studies have explored the organization and function of actin filaments on the target cancer cell side of the immunological synapse. However, there is growing evidence that the actin cytoskeleton of cancer cells also undergoes extensive remodeling upon cytotoxic lymphocyte attack, and that such remodeling can alter physical and functional interactions at the immunological synapse. In this article, we review the current knowledge of actin organization and functions at both sides of the immunological synapse between cytotoxic lymphocytes and cancer cells, with particular focus on synapse formation, signaling and cytolytic activity, and immune evasion.

## 1. Introduction

The tumor microenvironment (TME) plays multiple and central roles in cancer progression, e.g., by promoting tumor invasion, chemo and radiation-resistance, and by modulating the antitumor immune response. Ongoing research in the field of tumor immunobiology has identified immune escape as a classical hallmark of cancer and characterized different escape strategies elaborated by cancer cells [1,2,3]. The immune cell population with the best studied anti-tumor effector functions are cytotoxic lymphocytes cells, including cytotoxic T lymphocytes (CTLs) and natural killer (NK) cells. The concept of immunoediting is highly dependent on these key cells of the adaptive and innate immune systems. That the immune system and cancer cells exist in a delicate balance was already described more than a hundred years ago by Paul Ehrlich, when he suggested that host defense may prevent neoplastic cells from developing into tumors [4]. Sir Frank Mac Farlane Burnet hypothesized later that tumor cell neo-antigens induce an immunological reaction against cancer and subsequently formulated the immune surveillance theory [5]. The elimination phase of the immunoediting process is characterized by successful immunosurveillance of cancer cells that are recognized and eradicated by cytotoxic lymphocytes [6]. Constant immune selection pressure allows tumor clones to emerge, which escape immune cell-mediated elimination in the so-called equilibrium phase. During this phase, tumor cells develop different stratagems to escape immune surveillance, such as altered expression of surface markers, immune cell inhibition and establishment of an immunosuppressive TME [7,8,9,10]. Finally, tumors progress into the third phase of the immunoediting process, the escape phase, which leads to faster disease progression and poorly immunogenic tumors. Tumor cells have the ability to change their surroundings to their benefit, a feature that is accentuated by the characteristics of the TME, such as hypoxia—as a result of poor vascularization and rapid proliferation of cancer cells [11,12,13]. Hypoxia creates a hostile environment for cytotoxic immune cells that hinders their activation and effectiveness, while promoting activity of immune suppressive cell populations [11,14,15,16,17].

The actin cytoskeleton mainly consists in a complex network of polarized actin filaments (AFs) that contributes to nearly all fundamental cellular processes, including morphogenesis, motility, differentiation, division, membrane trafficking and signaling and the reader is invited to read the following review suggestions referring to these fields [18,19,20,21,22,23,24,25]. The actin cytoskeleton is subject to the activity of over 100 actin-binding proteins (ABPs) that regulate the organization and dynamics of AFs [26,27]. Broadly, ABPs can be distinguished according to their functions as actin nucleators, AF severing and capping proteins, and AF crosslinkers [25,26]. Actin polymerization occurs by polymerization of globular actin monomers, a process facilitated by actin nucleators, such as the Arp2/3 complex and formins, which promote the assembly of branched and linear arrays of AFs, respectively [26,28,29,30]. Further organization of AFs into higher-order structures, such as parallel bundles and three-dimensional networks, is mediated by crosslinking proteins of differing structural properties [31]. Severing proteins, such as actin depolymerizing factors (ADFs) and cofilin, play important roles in regulating AF dynamics by either increasing the amount of fast-growing barbed ends available for polymerization or by accelerating depolymerization from AF pointed ends [26,32].

A fundamental process underlying cytotoxic lymphocyte-mediated cancer cell killing is the formation of a specialized cell-cell junction, referred to as the immunological synapse (IS), between the immune cell and its prospective target. Different types of ISs can be formed, postulated to execute several functions including but not limited to the directed secretion of cytokines or lytic granules in a lytic IS with the goal of target cell lysis. The formation of a lytic IS requires activation of CTLs and NK cells through interaction of the T cell receptor (TCR) with peptide-loaded major histocompatibility complex-I (pMHC-I) on the target cell surface, or recognition of increased stress-induced ligands or loss of inhibitory MHC-I on the target cell surface, respectively. Initiation, formation and maturation of a lytic IS between cytotoxic lymphocytes and targets is a delicate process associated with major rearrangements of the actin cytoskeleton in cytotoxic lymphocytes [33,34,35]. In addition, the directed secretion of cytolytic granules containing perforin and granzymes, which ultimately promotes osmotic cell death and caspase-mediated apoptosis, also critically rely on actin cytoskeleton remodeling on the immune cell side of the IS [36,37,38,39,40]. On healthy cells, self-antigens presented on the MHC-I are not recognized by the TCR and the amount of stress-induced ligands is below the threshold required for NK cell activation, resulting in the formation of an inhibitory IS that protects the target cell from lymphocyte-mediated cytotoxicity. Another function executed by the IS formed between CTLs and dendritic cells (DCs) is the regulation of lymphocyte activation and maturation. A similar type of IS has been reported for the cell-cell-contact between NK cells and mature DCs and this interaction was termed regulatory IS [35,41,42,43]. Intercellular communication in a regulatory synapse can induce CTL activation and priming, increase the capacity of NK cells to release cytokines, and promote DC maturation. The different types of ISs are depicted in Figure 1.

Specific changes in actin cytoskeleton organization and dynamics are critically required for the initiation and stabilization of an IS, as well as for the activation and efficacy of associated immune cell effector functions. Such changes are the subject of much ongoing research [35]. Beside the well-defined actin cytoskeleton organization and roles in cytotoxic lymphocytes cells during IS formation and killing activity, there is increasing evidence that the actin cytoskeleton of cancer cells is also quickly remodeled in response to immune cell attack [44]. This review aims to discuss the role of the actin cytoskeleton in both, immune cells and cancer cells, during their interaction in an IS. The formation of an IS follows well-orchestrated spatiotemporal rearrangements of cytoskeletal components, that we will describe in detail. Particular attention will be paid to the organization and functions of the actin cytoskeleton during IS formation, signaling, and activity. In addition, we will discuss recent studies supporting that actin cytoskeleton remodeling in cancer cells can alter IS activity and lead to cancer immune evasion.

## 2. Actin Cytoskeleton and Immunological Synapse Architecture

### 2.1. Actin Assemblies at the Immunological Synapse

In CTLs, immune cell activation is triggered by interaction of the T cell receptor (TCR) with a non-self-antigen presented on the major histocompatibility complex class I (MHC-I) and engagement of further costimulatory receptors trigger [46,47,48,49]. In contrast, NK cell activity is regulated by the balance between activating and inhibiting ligands presented on the target cells surface. A vast array of inhibitory and activating receptors are co-expressed on the surface of NK cells, and activation occurs when signals from activating receptors dominate over the counterbalancing signals from inhibitory receptors [50,51]. In both CTLs and NK cells, engagement of activating receptors induces phosphorylation of membrane proximal signaling molecules and formation of a signalosome that includes many signaling and adapter molecules. Sustained signaling stimulates actin polymerization and generation of a branched AF network at the synapse periphery. The architecture of an IS is commonly divided in three regions defined by distinct actin assemblies and with specific functions in relation to signaling, adhesion, and cytolytic activities. The so-called supramolecular activation clusters (SMAC) are organized in a bulls-eye shape with a central SMAC (cSMAC), a peripheral SMAC (pSMAC), and a distal SMAC (dSMAC) from the inside out [35,52,53]. Live imaging of AF assembly and organization using fluorescent reporter, such as the small actin peptide Lifeact or GFP-actin, revealed a radially symmetric actin organization with differently assembled structures [54,55,56,57,58].

Initial engagement of the TCR–MHC-I complex activates actin polymerization in CTLs leading to a ring-shaped branched actin network in the dSMAC, shortly after contact with the target cell [48,59,60,61]. This Arp2/3-complex-mediated branched actin network, which exhibits high similarities to that of the lamellipodium of migrating cells, allows CTLs to spread across the surface of their targets [56,62], and provides support for the symmetric retrograde actin flow towards the cSMAC (see below). The pSMAC is composed of a lamella-like actin arc network that defines the radial symmetric contractile capability of this part of the IS [63]. Mechanistic studies have established that the assembly of the pSMAC actin arc is mediated by formins and organized by myosin II into antiparallel concentric arcs [63,64]. Indeed, selective depletion of the formin mDia prevents formation of the linear AF arrays required to generate the typical concentric myosin IIA-rich arcs [63,65,66]. The innermost part of the IS, the cSMAC, is referred to as an actin poor or actin hypodense region, which contains a fine AF network, which can only be visualized using super resolution imaging [53,66,67,68,69]. Moreover, the surface of CTLs it not flat, but is dominated by highly flexible and dynamic microvilli that show a high degree of AF polymerization at their core [70]. A recent study showed that using super resolution fluorescence imaging techniques and subsequent 3D reconstruction of the CTL membrane topography makes the tips of these membrane protrusions enriched for TCR clusters, which facilitate antigen screening. In addition, AF depolymerization not only inhibits microvilli formation, but also prevents the selective localization of TCR clusters [71]. Moreover, intercellular communication between CTLs and APCs is thought to be facilitated in parts by large, TCR-enriched membrane particles that are generated at the tip of these microvilli [72]. Delivery of these CTL-derived particles to cognate APCs was reported to increase DC activation status. Assembly of TCR microclusters was suggested to be a sequential process to which the actin cytoskeleton contributes by lowering the dissociation rates due to increased pulling forces at the IS, which maintain the different parts of the TCR signalosome together [73]. Intact actin dynamics may accordingly not only be involved in the retrograde movement of surface proteins towards the cSMAC but could also account for the observation that TCR microclusters are segregated from other protein clusters. Maintenance of the TCR signalosome composition during the lateral movement within the immune cell membrane is also subject to actin dynamics [74,75].

The integration of several different actin regulatory pathways underlying the formation of a functional IS makes this structure highly vulnerable to cytoskeletal alterations. The majority of actin cytoskeleton defects leading to primary immunodeficiencies (PIDs) are associated with a disruption of assembly and disassembly of AFs in multiple immune cell subtypes altering cell migration and adhesion, intercellular communication, intracellular signaling, and IS formation [76,77,78,79,80,81]. The architecture of the IS is subject to the spatiotemporal distinct activity of different actin polymerizing and regulatory proteins that result in topographical differences in AF organization. Engagement of activating receptors triggers Arp2/3 complex-dependent polymerization of branched AF, as illustrated in Figure 1. Polymerization activity of the Arp2/3 complex is regulated by different actin nucleation promoting factors (NPFs), such as WASp or WASp-verprolin homolog 2 (WAVE2). Actin regulation mediated by these NPFs does not occur on the effector immune cell side of the IS, but also plays an important role in DCs. Activity of WASp in target cells increases the overall IS area, and restricted knockdown of WASp in target DCs affects IS stability by inhibiting integrin polarization [82]. Actin cytoskeleton remodeling in target cells consequently represents a key component of normal IS formation through association of adhesive interaction and modulation of immunological synapse stability. These NPFs have to be associated with distinct functions corresponding to their localization within the IS. WAVE2 has been found primarily at the edges of the IS, and its activity has been associated with CTL adhesion and spreading over the antigen-presenting surface [83,84]. In addition to its role in modulation of IS stability, WASp is involved in the formation of actin-rich membrane protrusions at the secretory area, which is consistent with its predominant localization at the center of the IS [85,86,87].

Gene-expression profiling of peripheral CTLs from chronic lymphocytic leukemia (CLL) patients revealed a set of dysregulated actin regulatory proteins that was upregulated after cell-cell contact with leukemia cells [88]. Interestingly, the same cytoskeletal defects could be reproduced with CTLs from healthy donors that were cocultured with patient-derived CLL cells, indicating that cancer cells can promote cytoskeletal changes in cytotoxic lymphocytes. Further evaluation of the underlying mechanism identified several upregulated inhibitory ligands on leukemic cells that compromised the activation of key actin regulators including the Ras homolog (Rho) GTPases RhoA and Rac1, while promoting Cdc42 activity [89,90]. The combination of reduced RhoA/Rac1 signaling and increased Cdc42 activation resulted in the negative regulation of integrin activity in CTLs and subsequent adhesion and motility defects. Interestingly, defective actin polymerization was not limited to the IS between CLL cells and CTLs, but was also observed between APCs and CTLs, indicating that the cytoskeletal modifications in CTLs preventing functional IS formation were sustained over an extended period of time. A possible explanation for these effects comes from an elegant study in which it was demonstrated that leukemia cell-mediated changes in Rho GTPase activation result in impaired integrin signaling in CTLs, and that these defects could be rescued by the administration of the immunomodulatory drug lenalidomide, which resulted in the normalization of Rho GTPase activation signaling [90,91].

The ability of tumor cells to have a lasting negative impact on actin cytoskeletal dynamics in CTLs by compromising regulation of Rho GTPases shows a potential Achilles´ heel of cytotoxic lymphocytes. A variety of actin cytoskeleton regulators are involved at different spatiotemporal points during the formation and activity of a lytic IS, some of them were mentioned above. Dysregulation of one or two of these proteins can have a significant impact on anti-tumor immunity by inhibiting IS formation or affecting related functions, such as directed degranulation.

### 2.2. Actin Cytoskeleton Organization and Integrin Signaling at the Immunological Synapse

Binding of cytotoxic lymphocytes to target cells and IS maturation are dependent on the interaction of integrins on both sides of the IS. Cytotoxicity and polarized degranulation rely on a tight connection between CTLs and their prospective target cell. The integrin lymphocyte function-associated antigen 1 (LFA-1) was recently reported to be organized in nanometer-scaled clusters around the secretory domain of cytotoxic CD8+ T lymphocytes to allow confined delivery of cytotoxic granules [64,92,93]. The pSMAC shows the highest density of integrins, with a further increased LFA-1 density at the pSMAC/cSMAC boundary [94,95,96]. This integrin-mediated adhesive ring around the cSMAC was also suggested to help to seal the directed degranulation of lytic vesicles in the direction of the target cell [97,98]. Additionally, the sheer forces generated by the retrograde actin flow support binding of LFA-1 to its ligand intercellular adhesion molecule 1 (ICAM-1) by inducing a conformational change to its high affinity form. Beside this adhesion function, ICAM-1-bound LFA-1 also serves as a costimulatory signal for CTL and NK cell activation [96,99,100]. Topographical positioning and clustering of LFA-1 at the IS was recently shown to be confined to interstices within the actin cytoskeleton meshwork and underlie the activity of the actin nucleator WASp in CTLs [101]. The mobility of ICAM-1 in turn influences the actin flow-dependent signaling and cytotoxic lymphocyte functions. The actin cytoskeleton of antigen-presenting cells (APCs) engaged in an IS with CTLs was reported to limit the mobility of ICAM-1 on the cell surface. Release of this restriction by actin depolymerization in APCs negatively influences LFA-1 activation on T cells and adhesion properties [102]. Resistance of non-small cell lung cancer cells to CTL-mediated lysis was associated with the overexpression of the actin cytoskeleton regulators scinderin and ephrin-1A [103]. Scinderin is an actin severing protein and its overexpression is associated with increased actin dynamics, while signals of the receptor tyrosine kinases ephrin-A1 are translated into increased actin dynamics via activation of the N-WASp–Arp2/3 complex-pathway [104,105]. Silencing these genes using RNA interference restores both actin dynamics and cancer cell susceptibility to CTL-mediated lysis, giving indirect evidence that altered actin dynamics in cancer cells promotes immune evasion [103]. Detailed analysis of the IS formed between resistant non-small cell lung cancer (NSCLC) cells and CTLs revealed significant structural changes with sparse areas of tight cell-cell junctions and a wide IS cleft, contrasting with the tight IS formed with susceptible NSCLC cells. In a follow up study, the focal adhesion kinase (FAK) signaling pathway and Rho GTPase signaling were found to play a role in NSCLC resistance to CTLs [106]. FAK is a non-receptor tyrosine kinase that is activated by phosphorylation in response to integrin signaling [107]. Activated FAK can be found where cells attach to extracellular matrix components and at cell-cell contact areas, where it plays an important role in regulating AF turnover [108]. Expression of FAK has been reported to be upregulated in certain types of cancer, including ovarian and breast cancer [109]. Experimental silencing of FAK in NSCLC induces both, adhesion deficiency and increased resistance to CTL-mediated lysis by impairing IS formation [103,106].

These studies highlight that adhesion of CTLs to their prospective target is a crucial prerequisite for effective target cell lysis. Actin cytoskeletal dynamics drive the segregation of transmembrane protein clusters, such as integrins and induce their maturation by exertion of sheer forces to further increase their ligand affinity. Cancer cells can exploit the dependence of integrins for immobilized ligands by increasing actin cytoskeletal dynamics and increased ligands motility to prevent formation of the sealing ring at the pSMAC/cSMAC border to reduce efficiency of cytotoxic granule delivery.

## 3. Actin Cytoskeleton and Immunological Synapse Signaling

### 3.1. Actin Cytoskeleton and Receptor Signaling

The majority of engaged TCRs assemble in microclusters with their associated costimulatory receptors in the so-called TCR signalosome and can be found within the pSMAC. Signalosomes can also be found in the NK cell IS, although their composition differs from the TCR signalosome [110,111]. From the periphery, signalosomes move following the retrograde actin flow until they reach the pSMAC/cSMAC boundary [56,64]. Treatment with actin depolymerizing drugs, such as latrunculin, prevents microcluster-formation and consequently CTL and NK cell activation, highlighting the importance of actin dynamics in signaling at the IS [58,95]. Robust signaling of TCR microclusters occurs during the transition from the dSMAC to the cSMAC, with an abrupt stop upon reaching of the cSMAC [58]. As part of the TCR signalosome, linker of activation of T cells (LAT) recruits other adaptor proteins such as Nck and Vav1 that integrate TCR signaling into the rearrangement of the actin cytoskeleton by modulating WASp activity [60,66,112]. In the actin hypodense cSMAC, the connection between the signalosome and the actin cytoskeleton is lost, and thus TCR signaling comes to an abrupt stop after transition of TCR microclusters into the cSMAC, where exhausted TCRs are internalized in a Rho-dependent manner [35,58,113]. The strength of TCR signaling is directly correlated to the time it takes TCR microclusters to transit through the dSMAC/pSMAC region of the IS and, in primary CTLs, this depends on the speed of the retrograde actin flow [63,65]. Integrin signaling slows the retrograde actin flow resulting in activating signal by the TCR over longer periods of time. The actin-based molecular motor myosin IIA is not only involved in CTL motility and spreading over an antigen-coated surface, but has also been suggested to take part in the formation of the pSMAC and cSMAC, maturation of the IS, and persistence of TCR signaling [114]. Formation of TCR microclusters and their retrograde flow were both associated with recruitment and sustained activity of myosin IIA [115]. However, the exact role and importance of myosin IIA in IS formation and TCR signaling remain a matter of debate [114,116].

Although the role of activating receptor microclusters in activation of cytotoxic lymphocytes has been well established, much less is known about the distribution and stabilization of activating and inhibitory ligands on target cells. Cytotoxic lymphocytes do not only form an IS with infected or cancer cells, but also with APCs, such as DCs, with the aim to promote immune functions on both sides of the IS. In this regard, recent studies have shown that DCs also exhibit prominent actin polymerization in the region of the IS upon conjugation with CTLs and NK cells [43,82]. Such cytoskeletal remodeling was also shown to stabilize inhibitory ligands at the IS, and thereby prevent activation of immune cell effector functions. Preventing actin cytoskeleton polymerization in DCs was sufficient to increase the release of cytokines and cytotoxicity of NK cells against conjugates DCs. Such an IS was termed “regulatory synapse” [45] and was also reported between CTLs and DCs [82,117]. All in all, these works support that the actin cytoskeleton plays an important role in bridging TCR signaling to AF polymerization and that actin polymerization in DCs are the basis for the strong inhibitory signals in a regulatory IS.

Classification of functionally different ISs was proposed based on the status of actin polymerization on both sides of the IS, including the inhibitory synapse with a healthy target cell (AF accumulation on either side), the lytic synapse with a transformed target cell (AF accumulation in the immune cell only), and the regulatory synapse between DC and cytotoxic immune cells (AF accumulation in the APC only, as seen in Figure 1) [45]. Recently, we provided evidence that cancer cells use a similar mechanism as the regulatory synapse to prevent the activation of interacting NK cells [44,118]. Using high resolution confocal microscopy and high-throughput techniques, we found that resistant breast cancer cells in conjugation with cytotoxic NK cells accumulate AFs at their side of the IS, while susceptible cells do not [44]. As observed with DCs [43], the inhibition of actin polymerization using RNA interference was sufficient to abolish cancer cell resistance to NK cell-mediated lysis [44]. Moreover, actin accumulation in resistant cancer cells was also associated with increased levels of inhibitory ligands, including PD-L1 and MHC-I, at the IS. Both types of ligands induce negative receptor signaling in NK cells and can tip the balance between activating and inhibitory signaling in favor of cancer cells. This suggests that resistant cancer cells can mimic an increased expression of inhibitory ligands and thereby escape killing by NK cells, and possibly, by CTLs (through PD-L1) [118]. Accordingly, we propose a fourth type of IS that is formed between NK cells (possibly CTLs) and resistant cancer cells, which is characterized by actin polymerization on the target cell side (Figure 1). We named it “evasion synapse”.

### 3.2. Actin Cytoskeleton and Mechanopotentiation of Signaling and Killing at the Immunological Synapse

Over the last years, it has become increasingly clear that the actin cytoskeleton is not only a platform, but actively involved in the integration of spatial, molecular, and biochemical signals, as well as mechanical features into the formation and maintenance of the IS [119,120,121,122]. The forces submitted in the context of the IS are the result of membrane dynamics, cell spreading, and rearrangement of the actomyosin cytoskeleton.

Recently, direct evidence was given that the TCR acts as a mechanoreceptor and that activation of TCR signaling only occurs during applied tangential force, indicating that TCR–MHC-I interaction alone is not sufficient for CTL activation [123,124]. Following its activation, the TCR induces force changes at the IS by triggering dynamic actin cytoskeleton reorganization that translates into biochemical signals. As a result of antigen recognition, integrin-mediated adhesion force increases, with a direct correlation between TCR specificity and the strength of adhesion forces [125,126,127]. Forces in the frame of the IS have been shown to influence affinity maturation of integrins and strengthening of LFA-1/ICAM-1 interaction at the IS [128,129]. In addition, it was demonstrated that the cytotoxicity of CTLs is directly correlated to the mechanical forces applied on the conjugated target cell [130]. Furthermore, the actin cytoskeleton plays a key role as a main driver of force generation at the IS with regard to the retrograde translocation of receptor clusters originating from pushing and pulling forces at the dSMAC and pSMAC, respectively. The previously mentioned integrin-mediated adhesive ring around the cSMAC region; helps to apply the actin-based pulling force that supports pore formation by perforin [64,130]. In turn, actin remodeling on the target cell side of the IS locally modifies membrane properties, e.g., by inducing a substantial increase in cellular stiffness that can alter CTL activation [120,121,126,131]. CTLs interact with a variety of different cells, including endothelial cells, APCs, and target cells. The viscoelastic properties of endothelial cells and APCs change in response to inflammatory conditions and these changes are associated with myosin activity and actin polymerization dynamics [132,133]. Interestingly, mature APCs exhibit a higher degree of actin filament reorganization and a polarization of AFs towards the IS region [134,135]. This actin rearrangement in APCs was found to be critical for functional IS formation and T cell priming. Furthermore, it has been reported that ex vivo activation, expansion, and differentiation of CTLs is dependent on substrate rigidity [136]. CTLs interacting with immobilized stimulatory ligands show increased activation and proliferation under conditions of variable substrate stiffness as a result of increased signaling by the mechanosensitive TCR and other receptors [119,136]. The increased actin accumulation observed in APCs is thus sufficient to directly influence mechanotransduction by stabilizing inhibitory ligands at the IS, but also important for the maintenance of TCR signaling by providing a rigid substrate [122]. Additionally, mechanical forces at the IS can potentiate CTL-mediated killing of target cells. A recent study reported, that actin-enriched membrane protrusion in the dSMAC and cSMAC of the IS facilitate directed delivery of cytotoxic granules while also inducing the physical deformation of the target cell membrane [87]. These protrusions were dependent on WASp and Arp2/3 complex activity at the central parts of the IS and were a crucial requirement for force exertion in the context of the IS and potentiated target cell lysis.

## 4. Actin Cytoskeleton and Immunological Synapse Cytolytic Activity

In addition to signaling function, the IS is also the place of directed secretion of cytolytic granules toward cancer cells. Following interaction with a target in a lytic IS, CTLs and NK cells both show dynein-mediated polarization of the microtubule organizing center (MTOC) towards the IS [137,138,139]. The docking of the MTOC to the plasma membrane at the boundary between the actin hypodense cSMAC and the secretory domain of the IS is required for intracellular trafficking of lytic granules to the secretory domain of the IS [97] as seen in Figure 1. The content of lytic granules induces target cell death by osmotic cell lysis or via activation of caspases or pro-apoptotic proteins [140,141]. The mechanical force necessary for MTOC reorganization towards the membrane has been reported to involve different actin regulatory proteins, in particular formin family members that bind to microtubule plus ends. Accordingly, depletion of FMNL-1 and mDIA-1 has been shown to result in irregular MTOC positioning at the lytic IS [142,143]. It was initially suggested that the cytolytic granules can passively pass through the loose mesh of cortical AFs in the cSMAC [67,68]. However, recent high-resolution imaging-based studies established that, while AF dynamics at the cSMAC are reduced as compared to the directional actin flow in the dSMAC and the pSMAC, a fine network of AFs at the cSMAC shows dynamics at nanoscale level [69,144].

In vitro studies using cover slips or planar lipid bilayers coated with activating ligands or antibodies mimicking the target cell surface showed that in NK cells and CTLs in interaction with an activating surface, results in increased AF network mesh size at the cSMAC. The actin network mesh size reaches its peak at the same time the highest value of directed degranulation is observed. This granule-permissive clearance of actin is conserved in lytic ISs of both, CTLs and NK cells, although their actin clearance kinetics differ. A recent study found that in an IS formed by primary CTLs, actin clearance at the cSMAC appeared earlier, reached its peak faster and was short-lived compared to actin meshwork depolymerization observed in primary NK cells [69]. Using lattice light-sheet imaging, it could be shown that the MTOC and lytic granules clustered around it were positioned at the center of the cSMAC [53,139]. Based on the above results, it has been proposed that the actin network at the cSMAC serves as a barrier for lytic granules, and that transitory AF depletion decreases actin network density to initiate degranulation events [139]. Additionally, actin dynamics and myosin contractility have been shown to support the fusion of lytic granules with the plasma membrane by providing the required membrane tension. Furthermore, while myosin IIA activity was shown to be dispensable for NK cell adhesion to target cells, it was reported to interact directly with lytic granules, and facilitate the approach to the cell membrane at the secretory domain of the cSMAC [145,146]. Accordingly, mutations in the heavy chain of non-muscle myosin IIA were reported to be associated with immunodeficiency syndromes characterized by decreased CTL-mediated cytotoxicity despite otherwise normal IS formation [147,148,149]. The approach of lytic granules to the CTL membrane is accompanied by coronin 1A-mediated deconstruction of AFs at the cSMAC [53]. Another recent study also showed that secretion of perforin was associated with the formation of actin-rich protrusions in the secretory domain of CTLs, additionally highlighting that the cSMAC is not simply an actin-poor region as previously suggested, but exhibits highly regulated actin dynamics [87]. The termination of degranulation events is also actin cytoskeleton-dependent, as replenishment of AFs at the cSMAC acts as barrier for further granule degranulation and results in retraction of the MTOC from its position close to the IS area [150]. Therefore, rapid actin cytoskeleton remodeling seems to be crucial for initiation, but also the termination of directed secretion. Treatment of CTLs with actin depolymerizing drugs, such as Latrunculin A, increases duration of lytic granule secretion, underlining the role of AFs in creating a physical barrier for granule release.

Although only a few studies have directly investigated the organization and functions of the actin cytoskeleton in cancer cells during cytotoxic lymphocyte attack, there is growing evidence that actin remodeling plays a central role in mediating tumor immune evasion by altering either IS formation and function [44,103,106,151]. As previously discussed, we recently reported a prominent accumulation of AFs on the tumor cell side of the IS between intrinsically resistant breast tumor cells and NK cells, and that such process was correlated to immune evasion [44]. Selective targeting of actin cytoskeleton remodeling in cancer cells by targeted downregulation of the ARP2/3 complex regulators Cdc42 or N-WASP was sufficient to substantially increase target cell susceptibility to NK cell-induced lysis, as well as to restore high levels of NK cell derived-granzyme B levels in target cells. Similar to the regulatory synapse between DCs and NK cells, key inhibitory ligands, including HLA-A,B,C and PD-L1, were observed to accumulate at the evasion synapse together with actin accumulation in resistant cancer cells (Figure 1). A central role for the actin cytoskeleton in tumor immune evasion is further supported by an immune escape screen that identified Cdc42 as a mediator of increased resistance to antigen-specific CTL-mediated cytotoxicity [152]. Importantly, the expression of a constitutively active form of Cdc42 in mouse fibrosarcoma and human colorectal cancer cells was sufficient to induce resistance against CTLs and NK cell cytotoxicity in vitro, and promoted tumor growth in different mouse models, while it did not affect cell proliferation in vitro. Finally, additional evidence that actin remodeling promotes tumor immune evasion comes from the multiple links between the epithelial-to-mesenchymal transition (EMT), a process involving extensive modification of actin cytoskeleton organization and dynamics [153], and acquisition of cancer cell resistance to cytotoxic lymphocytes [154,155,156,157,158]. Such links have been recently reviewed and the reader is invited to refer to the following article [3].

## 5. Conclusions

Actin remodeling plays central roles in the three main processes underlying cytotoxic lymphocyte-mediated tumor cell killing, including target cell recognition, immune cell activation, and cancer cell killing. Actin remodeling at the cancer cell side of IS is emerging as an important mechanism of tumor immune evasion; however, our knowledge regarding this aspect remains fragmented. Additional work is required to identify clinically relevant targets to selectively impair actin organization and/or the dynamics in tumor cells and thereby restore an effective anti-tumor immune response. If and how the actin cytoskeleton of cancer cells could serve as a drug target for immunotherapy should be further investigated.

## Figures and Tables

**Figure 1 cells-08-00463-f001:**
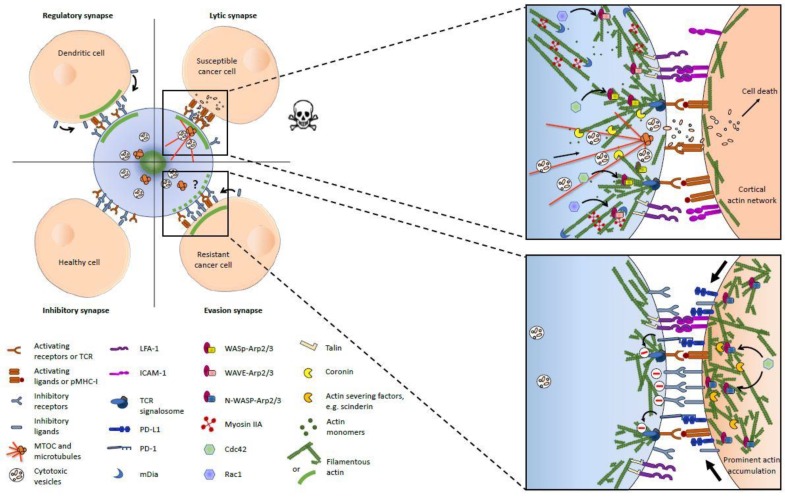
Actin cytoskeleton at the immunological synapses between a cytotoxic lymphocyte (blue) and its target cells. The left panel provides a general overview of the four types of immunological synapses (ISs) discussed in the article and shows the presence or absence of F-actin accumulation (large green line) on both sides of the IS (inspired from [45]). The specific actin filaments (AF) organization and some key upstream regulators, as well as the functions of AFs in intercellular signaling are depicted in the right panels for the lytic (upper panel) and evasion (lower panel) synapses. The reader should pay attention that the provided information sometimes originates exclusively from studies based on either natural killer (NK) cells or cytotoxic T lymphocytes (CTLs) and is invited to refer to the main text for more details.

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
