# Peer review of "Actin Cytoskeleton Straddling the Immunological Synapse between Cytotoxic Lymphocytes and Cancer Cells"

_cells, 2019, doi:10.3390/cells8050463_

Round 1

Reviewer 1 Report

Wurzer et al. here review actin dynamics during lymphocyte - tumour cell interactions; a highly timely topic that is currently generating a lot of interest in the field.

Overall the text is well organised and written, and easy to read. A particular strength of this review is that it covers actin dynamics on the tumour cell side of the synapse, which is usually ignored. Because of the dearth of work summarising these events, a figure or two, for instance conveying the different actin dynamics at the two sides of the synapse, would really help illustrate the comprehensive system.

The authors initially refer to the IS, which is described as an interface between a cytotoxic lymphocyte and a cancer cell. However, an IS can be any interface between a lymphocyte and an APC. Later in the text the authors distinguish between a lytic, inhibitory and and regulatory synapse, which should probably be defined right off the bat when introducing the IS, and care should be taken in how generically this concept is used. Similarly, when the structure or dynamics of actin during lysis is discussed, it may be worth distinguishing differences in the actin network between CTLs and NK cells, such as the degree of actin clearance, as these have recently been reported. 

For a review on actin dynamics at the synapse, myosin II is conspicuously under-represented. The actomyosin cortex is predominately responsible for the mechanical properties of both lymphocytes and cancer cells. Myosin motors have been shown to affect actin distribution, force generation, lytic potential of lymphocytes - these works need to be discussed.

In terms of original studies referenced, most are timely and relevant (however in the last year there have been studies on actin and WASP in CTLs from the Huse and Dupré labs, which should for the sake of completeness be included). The major issue is the number of other reviews referenced, please ensure that primary literature is cited instead where feasible.

Author Response

Dear reviewer,

We thank you for your comments on our manuscript entitled “Actin cytoskeleton straddling the immune synapse between cytotoxic lymphocytes and cancer cells”. The offered suggestions have been immensely helpfulfor revising and improving our paper. The responses to the comments are provided below.

1) […] Because of the dearth of work summarising these events, a figure or two, for instance conveying the different actin dynamics at the two sides of the synapse, would really help illustrate the comprehensive system.

Reply: In the revised version we have included a figure to help the reader better understand the actin cytoskeleton dynamics occurring on both sides of the immune synapse.

2) The authors initially refer to the IS, which is described as an interface between a cytotoxic lymphocyte and a cancer cell. However, an IS can be any interface between a lymphocyte and an APC. Later in the text the authors distinguish between a lytic, inhibitory and and regulatory synapse, which should probably be defined right off the bat when introducing the IS, and care should be taken in how generically this concept is used.Similarly, when the structure or dynamics of actin during lysis is discussed, it may be worth distinguishing differences in the actin network between CTLs and NK cells, such as the degree of actin clearance, as these have recently been reported. 

Reply: We agree with the reviewer and reformed the introduction as suggested. The distinct types of immune synapses are now shortly addressed following the introduction of the immune synapse and include differences between immune synapses formed by T lymphocytes and natural killer cells. Also, we now provide in section 4 a brief summary about the differences in actin clearance dynamics in cytotoxic T cells and natural killer cell immune synapses. We also clarified that the actin cytoskeleton dynamics in the context of an immune synapse are conserved between T cells and NK cells, with distinct differences in timing and duration.

3) For a review on actin dynamics at the synapse, myosin II is conspicuously under-represented. The actomyosin cortex is predominately responsible for the mechanical properties of both lymphocytes and cancer cells. Myosin motors have been shown to affect actin distribution, force generation, lytic potential of lymphocytes - these works need to be discussed.

Reply: During the revision of the manuscript, we included segments referring to myosin IIA activity. As suggested we now highlight the role of myosin in TCR signaling and actin cytoskeleton remodeling, force exertion, and cytolytic activity at the immune synapse. 

4) In terms of original studies referenced, most are timely and relevant (however in the last year there have been studies on actin and WASP in CTLs from the Huse and Dupré labs, which should for the sake of completeness be included). The major issue is the number of other reviews referenced, please ensure that primary literature is cited instead where feasible.

Reply: We exchanged review citations with references to primary research papers as far as possible. Also, we included further original work from the suggested work groups and beyond.

We would like to express our great appreciation to you and the work involved in the comments on our paper. If you have any further queries, please do not hesitate to contact us.

Kind regards, 

Clément Thomas.

Reviewer 2 Report

This review focuses on the role of actin cytoskeleton dynamics at the immune synapse formed between cytotoxic lymphocytes and target cancer cells in regulating signaling, immune effector functions and evasion. Overall the review is very interesting and comprises most of the literature in the field, highlighting the main findings.

I suggest the following points that could facilitate reading of the manuscript:

1-In the section corresponding to Actin Cytoskeleton and Synapse Architecture there is a large amount of information and molecules that are mentioned and/or briefly described. This section should be sub-divided into 2.1 Actin organization by regulatory molecules  and 2.2 Actin organization and integrins 

2-In line 165: NSCLC cells are not defined.

3-Brief concluding remarks should be included at the end of section 2 (line 174) and particularly in section 3 (line 213) where the relationship between actin remodeling and signaling should be summarized as well as the associated mechanisms that remain unaddressed. 

4-In section 3, the description of mechanical forces exerted at the IS and regulated by actin cytoskeleton should be included as a sub-section.

5-Finally, a simplified scheme of the main findings highlighted in this manuscript could allow readers to identify and differentiate what is known about the role of actin cytoskeleton dynamics or remodeling at the side of cytotoxic lymphocytes and the cancer cell.

Author Response

Dear reviewer,

We thank you for your comments on our manuscript entitled “Actin cytoskeleton straddling the immune synapse between cytotoxic lymphocytes and cancer cells”. We have carefully revised the manuscript according to the reviewers comments and provide point-by-point responses as follows.

1) In the section corresponding to Actin Cytoskeleton and Synapse Architecture there is a large amount of information and molecules that are mentioned and/or briefly described. This section should be sub-divided into 2.1 Actin organization by regulatory molecules and 2.2 Actin organization and integrins.

As suggested by the reviewer, we separated the chapter into the subsections “2.1 Actin assemblies at the immunological synapse” and “2.2 Actin cytoskeleton organization and integrin signaling at the immunological synapse”. We further included details on positioning of ligands with respect to actin cytoskeletal organization and downstream signaling affecting actin polymerization dynamics.

2) In line 165: NSCLC cells are not defined.

We apologize for this omission. We now present both the spelled-out version and the abbreviation. 

3) Brief concluding remarks should be included at the end of section 2 (line 174) and particularly in section 3 (line 213) where the relationship between actin remodeling and signaling should be summarized as well as the associated mechanisms that remain unaddressed.

We appreciate the valuable suggestions provided by the reviewer and refined the manuscript accordingly. As suggested, we have added conclusions to the end of both sections to summarize the connection between actin remodeling and signaling or force exertion, respectively.

4) In section 3, the description of mechanical forces exerted at the IS and regulated by actin cytoskeleton should be included as a sub-section.

As for comment 1), we separated the section in two subcategories for better understanding. As suggested, we summarized receptor mediated actin cytoskeleton reorganization in section “3.1 Actin cytoskeleton and receptor signaling”. Further, we included a subsection “3.2 Actin cytoskeleton and mechanopotentiation of signaling and killing at the immunological synapse” in which we address forces exerted by the actin cytoskeleton at the immune synapse and their impact and cytolytic efficiency.

5) Finally, a simplified scheme of the main findings highlighted in this manuscript could allow readers to identify and differentiate what is known about the role of actin cytoskeleton dynamics or remodeling at the side of cytotoxic lymphocytes and the cancer cell.

We included one figure to help the reader better understand the actin cytoskeleton dynamics occurring on both sides of the immune synapse.

We would like to express our great appreciation to you and the work involved in the comments on our paper. If you have any further queries, please do not hesitate to contact us.

Kind regards, 

Clément Thomas. 

Reviewer 3 Report

This is well organized and well written review paper. Authors reviewed well about the current concept of actin organization during immunological synapse formation between T cells and target cancer cells.

I only have a minor suggestion for the part of 3. Actin cytoskeleton and IS signaling.

Although many previous reports demonstrated that TCR microclusters are newly formed during T cell spreading, Jung et al (PNAS, 2016, E5916–E5924) recently demonstrated that TCR clusters are already localized at the microvilli tips on T cell surface before activation. In addition, Kim et al (Nature com, 2018, 9, 3630) showed that TCR-enriched microvilli particles are separated from T cell body and released during T cell kinapses. Interestingly, they also found that actin depolymerizing drug Latruculin (Lat) significantly remove the microvilli on the surface of T cells. Their findings and concepts must be discussed in this section. Somehow, Lat may not prevent the clusters but remove the microvilli from T cell surface.

Author Response

Dear reviewer,

We thank you for your comments on our manuscript entitled “Actin cytoskeleton straddling the immune synapse between cytotoxic lymphocytes and cancer cells”. We appreciate the helpful suggestions and revised the manuscript accordingly.

1) Although many previous reports demonstrated that TCR microclusters are newly formed during T cell spreading, Jung et al (PNAS, 2016, E5916–E5924) recently demonstrated that TCR clusters are already localized at the microvilli tips on T cell surface before activation. In addition, Kim et al (Nature com, 2018, 9, 3630) showed that TCR-enriched microvilli particles are separated from T cell body and released during T cell kinapses. Interestingly, they also found that actin depolymerizing drug Latruculin (Lat) significantly remove the microvilli on the surface of T cells. Their findings and concepts must be discussed in this section. Somehow, Lat may not prevent the clusters but remove the microvilli from T cell surface.

We have clarified that the topography of CTLs in an immune synapse is not flat but characterized by microvilli that show differences with regards to receptor distribution and clustering compared to the cell body. We also discussed the implications of these actin-based structures in intercellular communication. As suggested by the reviewer, both working concepts on receptor segregation and intercellular communication originating from CTL microvilli are now discussed in the review in line 135 – 150.

We would like to express our great appreciation to you and the work involved in the comments on our paper. If you have any further queries, please do not hesitate to contact us.

Kind regards, 

Clément Thomas.